# Using machine learning to predict risk of incident opioid use disorder among fee-for-service Medicare beneficiaries: A prognostic study

Wei-Hsuan Lo-Ciganic[1,2]*, James L. Huang[1,2], Hao H. Zhang[3], Jeremy C. Weiss[4], C. Kent Kwoh[5,6], Julie M. Donohue[7,8], Adam J. Gordon[9,10], Gerald Cochran[9], Daniel C. Malone[11], Courtney C. Kuza[8], Walid F. Gellad[8,12,13]

1 Department of Pharmaceutical Outcomes & Policy, College of Pharmacy, University of Florida, Gainesville, Florida, United States of America, 2 Center for Drug Evaluation and Safety (CoDES), College of Pharmacy, University of Florida, Gainesville, Florida, United States of America, 3 Department of Mathematics, University of Arizona, Tucson, Arizona, United States of America, 4 Carnegie Mellon University, Heinz College, Pittsburgh, Pennsylvania, United States of America, 5 Division of Rheumatology, Department of Medicine, University of Arizona, Tucson, Arizona, United States of America, 6 The University of Arizona Arthritis Center, University of Arizona, Tucson, Arizona, United States of America, 7 Department of Health Policy and Management, Graduate School of Public Health, University of Pittsburgh, Pittsburgh, Pennsylvania, United States of America, 8 Center for Pharmaceutical Policy and Prescribing, Health Policy Institute, University of Pittsburgh, Pittsburgh, Pennsylvania, United States of America, 9 Division of Epidemiology, Department of Internal Medicine, Program for Addiction Research, Clinical Care, Knowledge, and Advocacy, University of Utah, Salt Lake City, Utah, United States of America, 10 Informatics, Decision-Enhancement, and Analytic Sciences Center, Salt Lake City VA Health Care System, Salt Lake City, Utah, United States of America, 11 Department of Pharmacotherapy, College of Pharmacy, University of Utah, Salt Lake City, Utah, United States of America, 12 Division of General Internal Medicine, School of Medicine, University of Pittsburgh, Pittsburgh, Pennsylvania, United Sates of America, 13 Center for Health Equity Research Promotion, Veterans Affairs Pittsburgh Healthcare System, Pittsburgh, Pennsylvania, United States of America

* wlociganic@cop.ufl.edu

**Data Availability Statement:** Data are available from the Centers for Medicare and Medicaid Services for a fee and under data use agreement

## Abstract

### Objective

To develop and validate a machine-learning algorithm to improve prediction of incident OUD diagnosis among Medicare beneficiaries with ≥1 opioid prescriptions.

### Methods

This prognostic study included 361,527 fee-for-service Medicare beneficiaries, without cancer, filling ≥1 opioid prescriptions from 2011–2016. We randomly divided beneficiaries into training, testing, and validation samples. We measured 269 potential predictors including socio-demographics, health status, patterns of opioid use, and provider-level and regional-level factors in 3-month periods, starting from three months before initiating opioids until development of OUD, loss of follow-up or end of 2016. The primary outcome was a recorded OUD diagnosis or initiating methadone or buprenorphine for OUD as proxy of incident OUD. We applied elastic net, random forests, gradient boosting machine, and deep neural network to predict OUD in the subsequent three months. We assessed prediction performance

provisions. Per the data use agreement, the relevant limited data sets cannot be made publicly available. The website's reference on how others may access the relevant data, in the same manner as it was accessed by the authors of this study, is https://www.resdac.org/cms-virtual-research-data-center-vrdc-faqs.

**Funding:** National Institute on Drug Abuse R01DA044985 Drs. Wei-Hsuan Lo-Ciganic, James L. Huang, Hao H. Zhang, C. Kent Kwoh, Julie M. Donohue, Adam J. Gordon, Gerald Cochran, Daniel C. Malone, Courtney C. Kuza, and Walid F. Gellad Pharmaceutical Research and Manufacturers of America Foundation N/A Dr. Wei-Hsuan Lo-Ciganic.

**Competing interests:** We have read the journal's policy and the authors of this manuscript have the following competing interests: Dr. Kwoh has received honoraria from AbbVie and EMD Serono and has provided consulting services for Astellas, Thusane, and Novartis, EMD Serono and Express Scripts. I confirm that this does not alter our adherence to PLOS ONE policies on sharing data and materials.

using C-statistics and other metrics (e.g., number needed to evaluate to identify an individual with OUD [NNE]). Beneficiaries were stratified into subgroups by risk-score decile.

## Results

The training (n = 120,474), testing (n = 120,556), and validation (n = 120,497) samples had similar characteristics (age ≥65 years = 81.1%; female = 61.3%; white = 83.5%; with disability eligibility = 25.5%; 1.5% had incident OUD). In the validation sample, the four approaches had similar prediction performances (C-statistic ranged from 0.874 to 0.882); elastic net required the fewest predictors (n = 48). Using the elastic net algorithm, individuals in the top decile of risk (15.8% [n = 19,047] of validation cohort) had a positive predictive value of 0.96%, negative predictive value of 99.7%, and NNE of 104. Nearly 70% of individuals with incident OUD were in the top two deciles (n = 37,078), having highest incident OUD (36 to 301 per 10,000 beneficiaries). Individuals in the bottom eight deciles (n = 83,419) had minimal incident OUD (3 to 28 per 10,000).

## Conclusions

Machine-learning algorithms improve risk prediction and risk stratification of incident OUD in Medicare beneficiaries.

## Introduction

In 2017, 11.8 million Americans reported misuse of prescription opioids, [1] and 2.1 million suffered from opioid use disorder (OUD). [2–4] Opioid overdose deaths quintupled from 1999 to 2017. Although the specific opiates involved have changed over time, [2] prescription opioids were still involved in over 35% of opioid overdose deaths in 2017. [5] Many individuals with heroin use (40%-86%) reported misuse or abuse of opioid prescriptions before initiating heroin. [6]

The ability to identify individuals at high risk of developing OUD may inform prescribing and monitoring of opioids and can have a major impact on the size and scope of intervention programs (e.g., outreach calls from case managers, naloxone distribution). [7–10] Methods for identifying 'high-risk' individuals vary from identifying those with various high opioid dosage cut-points to the number of pharmacies or prescribers a patient visits. [11, 12] For example, Medicare uses these simple criteria to select which beneficiaries are enrolled into Comprehensive Addiction and Recovery Act (CARA) Drug Management Programs. [13] However, a recent study indicated that the Centers for Medicare & Medicaid Services (CMS) opioid high-risk measures miss over 90% of individuals with an actual OUD diagnosis or overdose. [14]

Several studies have developed automated algorithms to identify nonmedical opioid use and OUD using claims or electronic health records. [15–30] These algorithms mainly use traditional statistical methods to identify risk factors but do not focus on predicting an individual's risk. [15–30] Single risk factors are not necessarily strong predictors. [31] Recent studies have highlighted the shortcomings of current OUD prediction tools and call for developing more advanced models to improve identification of individuals at risk (or no risk) of OUD. [14, 26, 32–34] In particular, use of machine-learning techniques may enhance the ability to handle numerous variables and complex interactions in large data and generate predictions that can be acted upon in clinical settings. [35–41]

We previously successfully developed a machine-learning algorithm in Medicare to predict risk of overdose that attained a C-statistic over 0.90. [41] Here, we extend that work to develop and validate a machine-learning algorithm to predict incident OUD among Medicare beneficiaries having at least one opioid prescription. We then stratify beneficiaries into subgroups with similar risks of developing OUD to support clinical decisions and to improve intervention targeting. We chose Medicare because it offers the availability of longitudinal national claims data with a high prevalence of prescription opioid use and because the recently passed SUPPORT Act requires all Medicare Part D plan sponsors to establish drug management programs for at risk beneficiaries for opioid-related morbidity by 2022. [8]

## Materials and methods

### Design and sample

This is a prognostic study with a retrospective cohort design. It was approved by the University of Arizona Institutional Review Board. We used the Standards for Reporting of Diagnostic Accuracy (STARD) and the Transparent Reporting of a Multivariable Prediction Model for Individual Prognostic or Diagnosis (TRIPOD) guidelines for reporting our work (S1 and S2 Appendices). [42, 43]

From a 5% random sample of Medicare beneficiaries between 2011 and 2016, [44] we included prescription drug and medical claims in our sample. We identified fee-for-service adult beneficiaries aged ≥18 years who were US residents and received ≥1 non-parenteral and non-cough/cold opioid prescriptions. An index date was defined as the date of a patient's first opioid prescription between 07/01/2011 and 09/30/2016. We excluded beneficiaries who: (1) had malignant cancer diagnoses (S1 Table), (2) received hospice, (3) were ever enrolled in Medicare Advantage due to lack of medical claims needed to measure key predictors, (4) had their first opioid prescription before 07/01/2011 or after 10/1/2016, (5) were not continuously enrolled during the six months before the first opioid prescription, (6) had a diagnosis of OUD, opioid overdose, other substance use disorders, or received methadone or buprenorphine for OUD before initiating opioids, or (7) were not enrolled for three months after the first opioid fill (S1 Fig). We excluded beneficiaries who had a diagnosis of other substance use disorders to avoid confounding, because some physicians may have used this diagnosis when a patient had OUD and another substance use disorder. Beneficiaries remained in the cohort once eligible, regardless of whether or not they continued to receive opioids, until they had an occurrence of outcomes of interest, or were censored because of death or disenrollment.

### Outcome variables

Similar to many claims-based analyses, [27–30] our primary outcome was recorded diagnosis of OUD (S2 Table) or initiation of methadone or buprenorphine for OUD as a proxy for OUD in the subsequent 3-month period. We identified methadone for OUD using procedure codes (H0020, J1230) in outpatient claims, and buprenorphine for OUD in the Prescription Drug Events (PDE) file by products with FDA-approved indications for OUD. [41] Our secondary outcome was a composite outcome of incident OUD (i.e., OUD diagnosis or methadone or buprenorphine initiation) or fatal or nonfatal opioid overdose (prescription opioids or other opioids, including heroin). Opioid overdose was identified from inpatient or emergency department (ED) settings as defined in our study (S2 and S3 Tables). [41, 45–48]

## Candidate predictors

We compiled 269 candidate predictors identified from prior literature (S4 Table). [15–25, 44, 48–58] We measured a series of candidate predictors including patterns of opioid use, and patient, provider, and regional factors that were measured at baseline (i.e., within the three months before the first opioid fill) and in every 3-month period after prescription opioid initiation. To be consistent with the literature and quarterly evaluation period commonly used by prescription drug monitoring programs and health plans, a 3-month period was chosen. [19, 44, 59] We updated the predictors measured in each 3-month period to predict the risk of incident OUD in the subsequent 3-month period to account for changes in predictors over time (S2 Fig). This time-updating approach for predicting OUD risk in the subsequent three months mimics active surveillance that a health system might conduct in real time. Sensitivity analyses using all historical information prior to each 3-month period yielded similar results and are not further presented. S4 Table includes a series of variables related to prescription opioid and relevant medication use described in our previous work. [41]

## Machine-learning approaches and prediction performance evaluation

Our primary goal was risk prediction for incident OUD, and our secondary goal was risk stratification (i.e., identifying subgroups at similar OUD risk). To accomplish the first goal, we randomly and equally divided the cohort into three samples: (1) training sample to develop algorithms, (2) testing sample to refine algorithms, and (3) validation sample to evaluate algorithms' prediction performance. We developed and tested prediction algorithms for incident OUD using four commonly-used machine-learning approaches: elastic net (EN), random forests (RF), gradient boosting machine (GBM), and deep neural network (DNN). In prior studies, these methods have consistently yielded the best prediction results. [41, 49, 50] The S1 Text describes the details for each of the machine-learning approaches we used. Beneficiaries may have multiple 3-month episodes until occurrence of incident OUD or a censored event. Sensitivity analyses were conducted using iterative patient-level random subsets (i.e., using one 3-month period with predictors measured to predict risk in the subsequent three months for each patient) from the validation data to ensure the robustness of our findings.

To assess discrimination performance (i.e., the extent to which patients predicted as high risk exhibit higher OUD rates compared to those predicted as low risk), we compared the C-statistics (0.7 to 0.8: good; >0.8: very good) and precision-recall curves [51] across different methods from the validation sample using the DeLong Test. [52] OUD events are rare outcomes and C-statistics do not incorporate information about outcome prevalence, thus we also report eight metrics of evaluation: (1) estimated rate of alerts, (2) negative likelihood ratio (NLR), (3) negative predictive value, (4) number needed to evaluate to identify one OUD (NNE), (5) positive likelihood ratio (PLR), (6) positive predictive value (PPV), (7) sensitivity, and (8) specificity, to thoroughly assess our prediction ability (S3 Fig). [53, 54] For the EN final model, we report beta coefficients and odds ratios (ORs). EN regularization does not provide an estimate of precision and therefore 95% confidence intervals (95%CI) were not provided. [55]

No single threshold of prediction probability is suitable for every purpose, so to compare performance across methods, we present these metrics at multiple levels of sensitivity and specificity (e.g. arbitrarily choosing 90% sensitivity). We also used the Youden index to identify the optimized prediction threshold that balances sensitivity and specificity in the training sample. [56] Based on the individual's predicted probability of incident OUD, we classified beneficiaries in the validation sample into subgroups based on decile of risk score, with the highest decile further split into three additional strata based on the top 1st, 2nd to 5th, and 6th to

$10^{th}$ percentiles to allow closer examination of patients at highest risk of developing OUD. Using calibration plots, we evaluated the extent to which the observed risks of a risk subgroup agreed with the group's predicted OUD risk by the risk subgroup.

To increase clinical utility, we conducted several additional analyses. First, while the primary clinical utility of our machine-learning algorithm is to create a prediction risk score for developing incident OUD, we report the top 25 important predictors to provide some insights on variables relevant for prediction. However, interpreting individual important predictors separately or for causal inference should be done cautiously. Second, we compared our prediction performance with any 2019 CMS opioid safety measures over a 12-month period. [57] These CMS measures, which are meant to identify high-risk individuals or utilization behavior in Medicare, include three metrics: (1) high-dose use, defined as >120 MME for ≥90 continuous days, (2) ≥4 opioid prescribers and ≥4 pharmacies, and (3) concurrent opioid and benzodiazepine use for ≥30 days. Third, we conducted sensitivity analyses by excluding individuals diagnosed with OUD during the first three months. Fourth, Part D plan sponsors might only have access to their beneficiaries' prescription claims that may be more immediately available for analysis than medical claims. We thus compared prediction performance using variables only available in PDE files to all variables in the medical claims and PDE files and other linked data sources.

## Statistical analysis

We compared our three (training, testing, and validation) samples' patient characteristics with analysis of variance, chi-square test, two-tailed Student's $t$-test, or corresponding nonparametric test, as appropriate. All analyses were performed using SAS 9.4 (SAS Institute Inc, Cary, NC), and Python v3.6 (Python Software Foundation, Delaware, USA).

## Results

### Patient characteristics

Beneficiaries in the training (n = 120,474), testing (n = 120,556), and validation (n = 120,497) samples had similar characteristics and outcome distributions (81% aged ≥65 years, 61% female, 84% white, 26% with disability status and 30% being dually eligible for Medicaid; Table 1). Overall, 5,555 beneficiaries (1.54%) developed OUD and 6,260 beneficiaries (1.7%) had an incident OUD or overdose diagnosis after initiating opioids during the study period. Beneficiaries were followed for an average of 11.0 quarters and a total of 3,969,834 observation episodes.

### Prediction performance across machine-learning methods

Fig 1 summarizes the four prediction performance measures of each model. At the episode level, the four machine-learning approaches had similar performance measures for predicting OUD (Fig 1A): DNN (C-statistic = 0.881, 95%CI = 0.874–0.887), GBM (C-statistic = 0.882, 95%CI = 0.875–0.888), EN (C-statistic = 0.880, 95%CI = 0.873–0.886), and RF (C-statistic = 0.874, 95%CI = 0.867–0.881). EN required the fewest predictors compared to other approaches (EN = 48 vs. DNN = 270, GBM = 169, and RF = 255). DNN had slightly better precision-recall performance (Fig 1B), based on the area under the curve. Sensitivity analyses using randomly and iteratively selected patient-level data overall yielded similar results (see S4A–S4D Fig for an example).

S5 Table shows the performance measures for predicting incident OUD across different levels (90%-100%) of sensitivity and specificity for each method. When set at the optimized

**Table 1. Development of opioid use disorder and sociodemographic characteristics among Medicare beneficiaries (n = 361,527), divided into training, testing, and validation samples.**

| Characteristic | Training (n = 120,474) n (% of sample) | Testing (n = 120,556) n (% of sample) | Validation (n = 120,497) n (%of sample) |
|---|---|---|---|
| Development of opioid use disorder | 1,844 (1.5) | 1,842 (1.5) | 1,869 (1.6) |
| Age ≥ 65 years | 97,673 (81.1) | 97,707 (81.1) | 97,788 (81.2) |
| Female | 73,933 (61.4) | 73,769 (61.2) | 73,842 (61.3) |
| Race | | | |
| White | 100,602 (83.5) | 100,687 (83.5) | 100,744 (83.6) |
| Black | 11,156 (9.3) | 11,168 (9.3) | 11,132 (9.2) |
| Other | 8,716 (7.2) | 8,701 (7.2) | 8,621 (7.2) |
| Disabled eligibility | 30,711 (25.5) | 30,668 (25.4) | 30,813 (25.6) |
| Medicaid dual eligible | 36,787 (30.5) | 36,845 (30.6) | 36,614 (30.4) |
| Medicare Part D Low income subsidy | 30,711 (25.5) | 30,668 (25.4) | 30,813 (25.6) |
| End stage renal disease | 36,787 (30.5) | 36,845 (30.6) | 36,614 (30.4) |
| County of residence | | | |
| Metropolitan | 91,337 (75.8) | 91,427 (75.8) | 91,556 (76.0) |
| Non-metropolitan | 29,137 (24.2) | 29,129 (24.2) | 28,941 (24.0) |

sensitivity and specificity as measured by the Youden index, EN had an 81.5% sensitivity, 78.5% specificity, 0.54% PPV, 99.9% NPV, NNE of 184, and 22 positive alerts per 100 beneficiaries; and GBM had an 80.4% sensitivity, 80.4% specificity, 0.59% PPV, 99.9% NPV, NNE of 170, and 20 positive alerts per 100 beneficiaries (Fig 1C and 1D; S5 Table). When the sensitivity

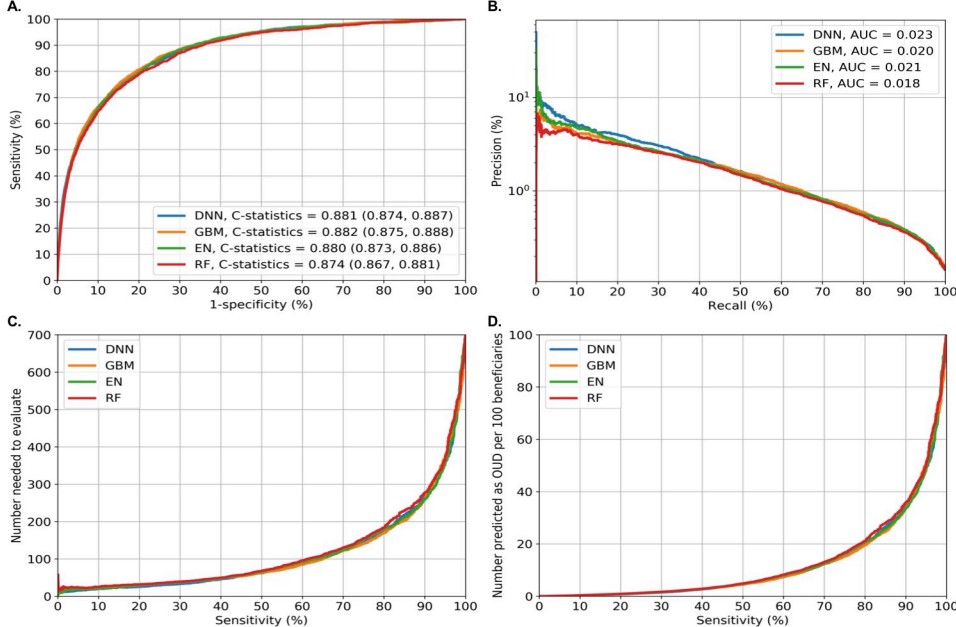

**Fig 1. Performance matrix across machine learning models for predicting incident opioid use disorder in Medicare beneficiaries.** Fig 1 shows four prediction performance matrices in the validation sample (120,497 beneficiaries with 1,298,189 non-OUD episodes and 1,869 OUD episodes). Fig 1A shows the areas under ROC curves (or C-statistics); Fig 1B shows the precision-recall curves (precision = PPV and recall = sensitivity): precision recall curves that are closer to the upper right corner or are above another method have improved performance; Fig 1C shows the number needed to evaluate by different cutoffs of sensitivity; and Fig 1D shows alerts per 100 patients by different cutoffs of sensitivity. Abbreviations: AUC: area under the curves; DNN: deep neural network; EN: elastic net; GBM: gradient boosting machine; RF: random forest; ROC: Receiver Operating Characteristics.

was instead set at 90% (i.e., attempting to identify 90% of individuals with an actual OUD), EN and GBM both had a 67% specificity, 0.39% PPV, 99.9% NPV, NNE of 259 to identify 1 individual with OUD, and 33 positive alerts generated per 100 beneficiaries (S5 Table). When, instead, specificity was set at 90% (i.e., identifying 90% of individuals with actual non-OUD), EN and GBM both had a ~66% sensitivity, ~0.95% PPV, 99.9% NPV, 106 NNE, and 10 positive alerts per 100 beneficiaries.

For the secondary outcome (i.e., combined incident OUD or overdose), DNN and GBM outperformed EN and RF (C-statistic: >0.87 vs. 0.86). GBM required fewer predictors than DNN (DNN = 268, GBM = 140; S5A–S5D Fig). When sensitivity was set at 90%, GBM had a 72% specificity, 0.57% PPV, 99.9% NPV, NNE of 177 to identify one individual with incident OUD or overdose, and 30 positive alerts generated per 100 beneficiaries (S6 Table). Other results are consistent with the findings for predicting incident OUD.

## Risk stratification by decile risk subgroup

Fig 2 depicts the actual OUD rate for individuals in each decile subgroup using EN. The high-risk subgroups (with risk scores in the top decile; 15.8% [n = 19,047] of the validation cohort) had a positive predictive value of 0.96%, a negative predictive value of 99.8%, and NNE of 104. Among all 360 individuals with incident OUD, 248 (69%) occurred in the top two decile subgroups (decile 1 = 50.8% and decile 2 = 18.1%). Those in the 1st decile subgroup had at least a 10-fold higher OUD rate compared to the lower-risk groups (e.g., observed OUD rate: decile 1 = 3.01%, decile 2 = 0.36%, decile 10 = 0.19%). The 3rd through 10th decile subgroups had minimal rates of incident OUD (3 to 28 per 10,000).

The EN and DNN's algorithms had high concordant prediction performance (S6 Fig). Fig 3 shows the 25 most important predictors identified by EN, including lower back pain, Elixhauser drug abuse indicator (excluding OUD), Schedule IV short-acting opioids (i.e., tramadol), disability as the reason for Medicare eligibility, and having urine drug tests. S7 Fig shows the top 25 important predictors (e.g., age, total MME, lower back pain) for incident OUD and incident OUD or overdose identified by the GBM model.

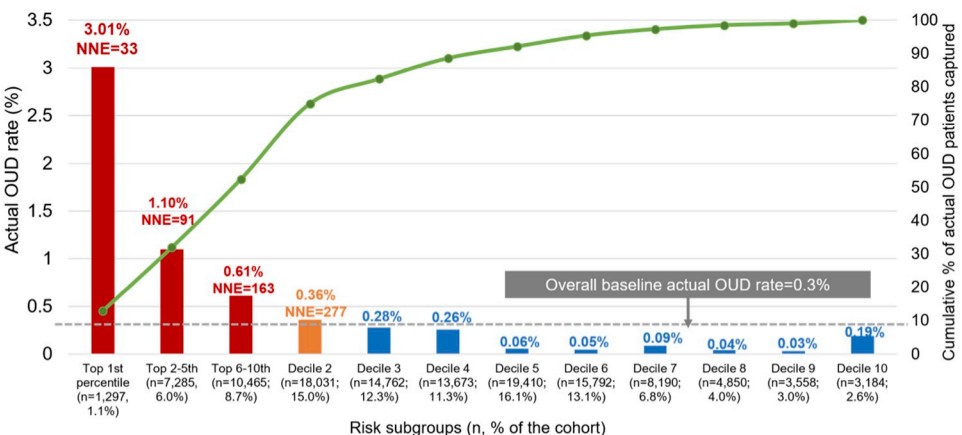

**Fig 2. Incident OUD identified by elastic net's decile risk subgroup in the validation sample (n = 120,497)[a].**
Abbreviation: OUD: opioid use disorder. [a]: Based on the individual's predicted probability of an OUD event, we classified beneficiaries in the validation sample into decile risk subgroups, with the highest decile further split into 3 additional strata based on the top 1, 2nd to 5th, and 6th to 10th percentiles to allow closer examination of patients at highest risk of developing OUD.

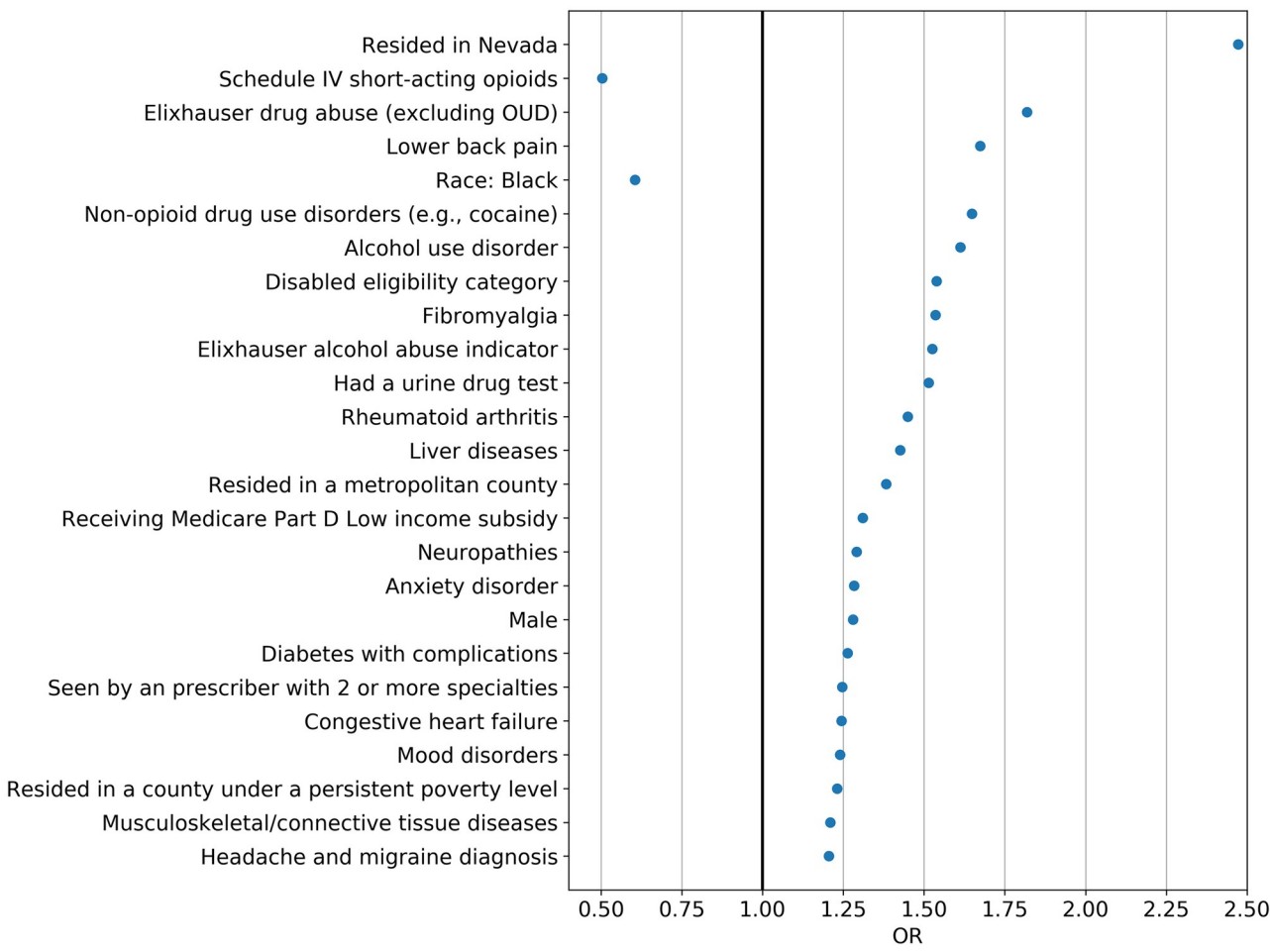

**Fig 3. Top 25 predictors for opioid use disorder identified by elastic net (ordered by importance)[a]. [a]**Figure shows the important predictors ordered by feature importance based on odds ratios. EN regularization does not provide an estimate of precision and therefore 95% confidence intervals (95% CI) were not provided. Abbreviations: OR: odds ratios; OUD: opioid use disorder.

## Secondary and sensitivity analyses

Table 2 compares EN's algorithms to use of any of CMS' opioid safety measures over a 12-month period. For example, by defining high risk as being in the top 5th percentiles of risk scores, EN captured 69% of all OUD cases (NNE = 29) over a 12-month period, compared to 27.3% using CMS measures. S7 Table presents the comparisons of the prediction performance for CMS high-risk opioid use measures with DNN and GBM over a 12-month period.

Sensitivity analyses excluding incident OUD occurring in the first three months had a similar performance with the main analyses (S8A–S8D Fig). Finally, models using only variables from the PDE files did not perform as well as models using the full set of variables (using EN for example: C-statistic = 0.821 vs. 0.880; NNE = 322 vs. 170; and positive alerts rate = 48 vs. 33 per 100 beneficiaries with sensitivity set at 90%; S9A–S9D Fig).

## Discussion

We developed machine-learning models that perform strongly to predict the risk of developing OUD using national Medicare data. All of the machine-learning approaches had excellent discrimination (C-statistic >0.87) for predicting OUD risk in the subsequent three months.

**Table 2. Comparison of prediction performance using any of the Centers for Medicaid & Medicaid Services (CMS) high-risk opioid use measures vs. elastic net in the validation sample (n = 114,253) over a 12-month period[a].**

| Risk subgroups (n, % of the cohort) | Any CMS measure[b] | | High risk in elastic net using different thresholds[c] | | |
|---|---|---|---|---|---|
| | Low risk (n = 110,171, 96.4%) | High risk (n = 4,082, 3.57%) | Top 1 percentile (n = 2,207, 1.93%) | Top 5th percentile (n = 11,633, 10.18%) | Top 10th percentile (n = 23,541, 20.6%) |
| Number of actual OUD (% of each subgroup) | 412 (0.4) | 155 (3.8) | 186 (8.4) | 391 (3.4) | 475 (2.0) |
| Number of actual non-OUD (% of each subgroup) | 109,759 (99.6) | 3,927 (96.2) | 2,021 (91.6) | 11,242 (96.6) | 23,066 (98.0) |
| NNE | 270 | 26 | 11 | 29 | 49 |
| % of all OUD over 12 months (n = 567) captured | 72.7 | 27.3 | 32.8 | 69.0 | 83.8 |

**Abbreviations: NNE**: number needed to evaluate; **OUD: opioid use disorder**

[a]: The CMS measures were based on a 12-month period rather than three months. To compare CMS measures, beneficiaries were thus required to have at least a 12-month period of follow up and the resulting sample size was smaller than the sample size in the main analysis. If classifying beneficiaries with *any* of the CMS high-risk opioid use measures as OUD, the remaining will be consider as non-OUD.

[b]: The 2019 CMS' opioid safety measures are meant to identify high-risk individuals or utilization behavior.[1] These measures include 3 metrics (1) high-dose use, defined as >120 MME for ≥90 continuous days, (2) ≥4 opioid prescribers and ≥4 pharmacies, or (3) concurrent opioid and benzodiazepine use ≥30 days.

[c]: For elastic net, we presented high-risk groups using different cutoff thresholds of prediction probability: individuals with (1) predicted probability in the top 1 percentile (0.95); (2) predicted probability in the top 5th percentile (0.77) or above; and (3) predicted probability in the top 10th percentile (0.61) or above. If classifying beneficiaries in the high-risk group of OUD, the remaining will be consider as non-OUD.

Elastic net (EN) was the preferred and parsimonious algorithm because it required only 48 predictors, which may reduce computational time. Given the low incidence of OUD in a 3-month period, PPV was low, as expected. [53] However, this algorithm was able to effectively segment the population into different risk groups based on predicted risk scores, with 70% of the sample having minimal OUD risk, and half of the individuals with OUD captured in the top decile group. Identifing such risk groups can be a valuable prospect for policy makers and payers who currently target interventions based on less accurate risk measures. [14]

We identified eight prior published opioid prediction models, each focusing on predicting a different aspect of OUD: six-month risk of diagnosis-based OUD using private insurance claims; [30] 12-month risk of having aberrant behaviors of opioid use after an initial pain clinic visit; [15] 12-month risk of diagnosis-based OUD using private insurance claims [19, 23] or claims data from a pharmacy benefit manager [29]; two-year risk of clinical-documented prob-lematic opioid use in electronic medical records (EMR) in a primary care setting; [24] and five-year risk of diagnosis-based OUD using EMR from a medical center [27] and using Rhode Island Medicaid data; [28] These studies had several key limitations, including measuring predictors at baseline rather than over time, using case-control designs that might not be able to calibrate well to population-level data with the true incidence rate of OUD, and having a C-statistic of up to 0.85 in non-case-control designs. [15, 24, 28, 29] Our study overcomes these limitations by using a population-based sample and is the first study, to our knowledge, that predicts more immediate OUD risk (in the subsequent 3-month period) as opposed to a year or longer time period.

With any prognostic prediction algorithm, the selection of probability threshold inevitably results in a tradeoff between sensitivity and specificity and also depends on the type of inter-ventions triggered by a positive alert. Resource intensive interventions (e.g., pharmacy lock-in programs or case management) may be preferred for individuals in the highest risk subgroup, whereas lower cost or low-risk interventions (e.g., naloxone distribution) [7] may be used for

those in the moderate risk subgroups (e.g., top 6th-10th percentiles of predicted scores). We proposed several potential thresholds (e.g., top 1st percentile of risk scores) for classifying patients at high risk of OUD, allowing those who implement the algorithm to determine the optimal thresholds for their intervention of interest. Regardless of the threshold selected, our risk-stratified approach can first exclude a large majority (>70%) of individuals with negligible or minimal OUD risk prescribed opioids. Since the incidence of OUD in the subsequent three months is low, the PPV was low among all the potential thresholds (<3% in the top 1 percentile of EN's predicted scores). However, given the seriousness of the consequences of OUD and overdose, identifying subgroups with different risk magnitudes may represent clinically actionable information.

Our predicted model and risk stratification strategies can be used to more efficiently determine whether a patient is at high risk of incident OUD compared to recent CMS measures. [14] The EN model predicting OUD and the model predicting a composite outcome of OUD and overdose could first exclude a large segment of the population with minimal risk of the outcome. While the CMS opioid safety measures use only prescription data, over 70% of incident OUD cases occurred among those not viewed as high risk. Furthermore, in our sensitivity analysis, the EN models that included only prescription data did not perform as well as those including medical claims (e.g., doubled NNE and increased 1.5 times the number of positive alerts). Nonetheless, given the policy importance of risk prediction in Medicare Part D, additional consideration should be given to the criteria used to identify high-risk individuals.

Our study has several limitations. First, the claims data does not capture patients obtaining opioids from non-medical settings or paying out of pocket. Second, although OUD is likely to be underdiagnosed, [58, 59] it is captured with high specificity in claims data, suggesting that PPV and risk may be underestimated. Third, laboratory results and socio-behavioral information are not captured in administrative billing data. Furthermore, our study used publicly available older data. Updating and refining the prediction algorithm on a regular basis (e.g., quarterly or yearly) is recommended as opioid-related policies and practices have changed over time. Finally, our prediction algorithms were derived from the fee-for-service Medicare population and thus may not generalize to individuals in other populations with different demographic profiles or enrolled in programs with different features including Medicare Advantage plans. The analysis was not pre-registered and the results should be considered exploratory.

In conclusion, our study illustrates the potential and feasibility of machine-learning OUD prediction models developed using routine administrative claims data available to payers. These models have excellent prediction performance and can be valuable tools to more efficiently and accurately identify individuals at high risk or with minimal risk of OUD.

## Supporting information

**S1 Appendix. Compliance to the 2015 Standards for Reporting Diagnostic Accuracy (STARD) checklist.**
(DOCX)

**S2 Appendix. Compliance to the 2015 Transparent Reporting of a Multivariable Prediction Model for Individual Prognosis Or Diagnosis (TRIPOD) checklist.**
(DOCX)

**S1 Text. Appendix methods: Machine learning approaches used in the study.**
(DOCX)

**S1 Table. Diagnosis codes for the exclusion of patients with malignant cancers based on the National Committee for Quality Assurance (NCQA)'s Opioid Measures in 2018 Healthcare Effectiveness Data and Information Set (HEDIS).**
(DOCX)

**S2 Table. Diagnosis codes for identifying opioid use disorder and opioid overdose.**
(DOCX)

**S3 Table. Other diagnosis codes used to identify the likelihood of opioid overdose.**
(DOCX)

**S4 Table. Summary of predictor candidates (n = 269) measured in 3-month windows for predicting incident opioid use disorder or opioid overdose.**
(DOCX)

**S5 Table. Prediction performance measures for predicting incident opioid use disorder, across different machine learning methods with varying sensitivity and specificity.**
(DOCX)

**S6 Table. Prediction performance measures for predicting incident opioid use disorder or opioid overdose, across different machine learning methods with varying sensitivity and specificity.**
(DOCX)

**S7 Table. Comparison of prediction performance using any Centers for Medicare & Medicaid Services (CMS) high-risk opioid use measures vs. Deep Neural Network (DNN) and Gradient Boosting Machine (GBM) in the Validation sample (n = 114,253) over a 12-month period.**
(DOCX)

**S1 Fig. Sample size flow chart of study cohort.**
(TIF)

**S2 Fig. Study design diagram.**
(TIF)

**S3 Fig. Classification matrix and definition of prediction performance metrics.** Saito T, Rehmsmeier M. The precision-recall plot is more informative than the ROC plot when evaluating binary classifiers on imbalanced datasets. PLoS One. 2015;10(3):e0118432. Epub 2015/03/05. doi: 10.1371/journal.pone.0118432. PubMed PMID: 25738806; PubMed Central PMCID: PMCPMC4349800. Romero-Brufau S, Huddleston JM, Escobar GJ, Liebow M. Why the C-statistic is not informative to evaluate early warning scores and what metrics to use. Crit Care. 2015;19:285. Epub 2015/08/14. doi: 10.1186/s13054-015-0999-1. PubMed PMID: 26268570; PubMed Central PMCID: PMCPMC4535737.
(TIF)

**S4 Fig. Prediction performance matrix across machine learning approaches in predicting risk of incident opioid use disorder in the subsequent 3 months: Sensitivity analyses using patient-level data.** Figure shows four prediction performance matrices using an example of using randomly and iteratively selected patient-level data (n = 50,000 [49,927 non-OUD and 73 OUD patients], excluding those who had an OUD from the first 3-month period) from the validation sample. S4A Fig shows the areas under ROC curves (or C-statistics); S4B Fig shows the precision-recall curves (precision = PPV and recall = sensitivity)—precision recall curves that are closer to the upper right corner or above the other method have improved

performance; S4C Fig shows the number needed to evaluate by different cutoffs of sensitivity; and S4D Fig shows alerts per 100 patients by different cutoffs of sensitivity. Abbreviations: AUC: area under the curves; DNN: deep neural network; EN: elastic net; GBM: gradient boosting machine; RF: random forest; ROC: Receiver Operating Characteristics.
(TIF)

**S5 Fig. Prediction performance matrix across machine learning approaches in predicting risk of incident opioid use disorder or overdose in the subsequent 3 months: Sensitivity analyses using patient-level data.** Figure shows four prediction performance matrices for predicting incident OUD or overdose in the subsequent three months at the episode level from the validation sample. S5A Fig shows the areas under ROC curves (or C-statistics); S5B Fig shows the precision-recall curves (precision = PPV and recall = sensitivity)—precision recall curves that are closer to the upper right corner or above the other method have improved performance; S5C Fig shows the number needed to evaluate by different cutoffs of sensitivity; and S5D Fig shows alerts per 100 patients by different cutoffs of sensitivity. Abbreviations: AUC: area under the curves; DNN: deep neural network; EN: elastic net; GBM: gradient boosting machine; OUD: opioid use disorder; RF: random forest; ROC: Receiver Operating Characteristics.
(TIF)

**S6 Fig. Scatter plot between Gradient Boosting Machine (GBM) and Elastic Net's prediction scores.**
(TIF)

**S7 Fig. Top 25 important predictors for incident OUD and incident OUD/overdose selected by gradient boosting machine (GBM).** Abbreviations: ED: emergency department; FFS: fee-for-service; GBM: gradient boosting machine; MME: morphine milligram equivalent; No: number of [a] Rather than p values or coefficients, the GBM reports the importance of predictor variables included in a model. Importance is a measure of each variable's cumulative contribution toward reducing square error, or heterogeneity within the subset, after the data set is sequentially split based on that variable. Thus, it is a reflection of a variable's impact on prediction. Absolute importance is then scaled to give relative importance, with a maximum importance of 100. For example, the top 5 important predictors identified from GBM included age, total cumulative MME, lower back pain, average MME prescribed by provider per patient, and averaged no. monthly non-opioid prescriptions.
(TIF)

**S8 Fig. Prediction performance matrix across machine learning approaches in predicting risk of opioid use disorder in the subsequent 3 months: Sensitivity analyses at episode level (excluding the incident OUD cases occurring in the first 3-month period).** Figure shows four prediction performance matrices excluding opioid disorder outcomes occurred in the first 3 months after the index date in the validation sample. S8A Fig shows the areas under ROC curves (or C-statistics); S8B Fig shows the precision-recall curves (precision = PPV and recall = sensitivity)—precision recall curves that are closer to the upper right corner or above the other method have improved performance; S8C Fig shows the number needed to evaluate by different cutoffs of sensitivity; and S8D Fig shows alerts per 100 patients by different cutoffs of sensitivity. Abbreviations: AUC: area under the curves; DNN: deep neural network; EN: elastic net; GBM: gradient boosting machine; RF: random forest; ROC: Receiver Operating Characteristics.
(TIF)

**S9 Fig. Prediction performance matrix across machine learning approaches in predicting risk of opioid use disorder in the subsequent 3 months: Using variables from part D events file only.** Figure shows four prediction performance matrices using only variables from Prescription Drug Events files in the validation sample. S9A Fig shows the areas under ROC curves (or C-statistics); S9B Fig shows the precision-recall curves (precision = PPV and recall = sensitivity)—precision recall curves that are closer to the upper right corner or above the other method have improved performance; S9C Fig shows the number needed to evaluate by different cutoffs of sensitivity; and S9D Fig shows alerts per 100 patients by different cutoffs of sensitivity. Abbreviations: AUC: area under the curves; DNN: deep neural network; EN: elastic net; GBM: gradient boosting machine; RF: random forest; ROC: Receiver Operating Characteristics.
(TIF)

## Acknowledgments

We thank Debbie L. Wilson, PhD (University of Florida) for providing editorial assistance in the preparation of this manuscript.

## Disclosure

The views presented here are those of the authors alone and do not necessarily represent the views of the Department of Veterans Affairs or the United States Government.

## Author Contributions

**Conceptualization:** Wei-Hsuan Lo-Ciganic, Walid F. Gellad.

**Data curation:** Wei-Hsuan Lo-Ciganic, James L. Huang.

**Formal analysis:** Wei-Hsuan Lo-Ciganic, James L. Huang, Hao H. Zhang, Walid F. Gellad.

**Funding acquisition:** Wei-Hsuan Lo-Ciganic.

**Investigation:** Wei-Hsuan Lo-Ciganic, James L. Huang, Hao H. Zhang, Jeremy C. Weiss, Walid F. Gellad.

**Methodology:** Wei-Hsuan Lo-Ciganic, James L. Huang, Hao H. Zhang, Jeremy C. Weiss, Walid F. Gellad.

**Project administration:** Wei-Hsuan Lo-Ciganic, Walid F. Gellad.

**Resources:** Wei-Hsuan Lo-Ciganic, James L. Huang, Hao H. Zhang, Jeremy C. Weiss, C. Kent Kwoh, Walid F. Gellad.

**Supervision:** Wei-Hsuan Lo-Ciganic, Walid F. Gellad.

**Validation:** Wei-Hsuan Lo-Ciganic, James L. Huang.

**Visualization:** Wei-Hsuan Lo-Ciganic, Walid F. Gellad.

**Writing – original draft:** Wei-Hsuan Lo-Ciganic.

**Writing – review & editing:** Wei-Hsuan Lo-Ciganic, James L. Huang, Hao H. Zhang, Jeremy C. Weiss, C. Kent Kwoh, Julie M. Donohue, Adam J. Gordon, Gerald Cochran, Daniel C. Malone, Courtney C. Kuza, Walid F. Gellad.

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
