## [Decision Letter · Decision Letter 0]

29 Apr 2020

PONE-D-20-09504

Using machine learning to predict risk of incident opioid use disorder among fee-for-service Medicare beneficiaries: a prognostic study

PLOS ONE

Dear Dr. Lo-Ciganic,

Thank you for submitting your manuscript to PLOS ONE. After careful consideration, we feel that it needs some minor revisions. Therefore, we invite you to submit a revised version of the manuscript that addresses the points raised during the review process.

We would appreciate receiving your revised manuscript by Jun 12 2020 11:59PM. To enhance the reproducibility of your results, we recommend that if applicable you deposit your laboratory protocols in protocols.io, where a protocol can be assigned its own identifier (DOI) such that it can be cited independently in the future. For instructions see: http://journals.plos.org/plosone/s/submission-guidelines#loc-laboratory-protocols

We look forward to receiving your revised manuscript.

Kind regards,

Kevin Lu, PhD

Academic Editor

PLOS ONE

Journal Requirements:

. When submitting your revision, we need you to address these additional requirements.

https://jamanetwork.com/journals/jamanetworkopen/fullarticle/2728625

https://onlinelibrary.wiley.com/doi/10.1002/pds.4864

In your revision ensure you cite all your sources (including your own works), and quote or rephrase any duplicated text outside the methods section. Further consideration is dependent on these concerns being addressed.

"I have read the journal's policy and the authors of this manuscript have the following

competing interests:

Dr. Kwoh has received honoraria from AbbVie and EMD Serono and has provided

consulting services for Astellas, Thusane, and Novartis, EMD Serono and Express

Scripts."

Reviewers' comments:

Reviewer's Responses to Questions

**Comments to the Author**

1. Is the manuscript technically sound, and do the data support the conclusions?

Reviewer #1: Yes

Reviewer #2: Yes

2. Has the statistical analysis been performed appropriately and rigorously? 

Reviewer #1: Yes

Reviewer #2: Yes

3. Have the authors made all data underlying the findings in their manuscript fully available?

Reviewer #1: Yes

Reviewer #2: Yes

4. Is the manuscript presented in an intelligible fashion and written in standard English?

Reviewer #1: Yes

Reviewer #2: Yes

5. Review Comments to the Author

Reviewer #1: Major comments:

The clinical utilization of the algorithm is not clear. The authors have identified 25 important predictors. However, it is not clear how to use these predictors in clinical practice.

Minor comments

Figure 3: What does “e.g.” indicate? Different states have different OR?

Reviewer #2: Page 4. Abstract. “CONCLUSIONS : Machine-learning algorithms improve risk prediction and stratification of incident OUD, especially in identifying low-risk subgroups that have negligible risk.”

The main objective of this study is to develop a machine-learning algorithm which could help identify patients at OUD risks, while the statement regarding the low-risk groups may not be directly related to the main purpose.

Page 15. Line 271. “Figure 3 shows the 25 most important predictors identified by EN, including lower back pain,”

It would be great to discuss the clinical implications for these predictors, and potential interventions. Patients with lower back pain are very likely to receive opioids, would that be considered as OUD?

Page 19. Line 337. “determine whether a patient is at high risk of incident OUD compared to current CMS”

The data for this study was up to 2016, is there any changes in the policy or physician’s practices? Would it be necessary to update the algorism using most recent data?

6. PLOS authors have the option to publish the peer review history of their article (what does this mean?). If published, this will include your full peer review and any attached files.

Reviewer #1: No

Reviewer #2: No

---

## [Author Response · Author response to Decision Letter 0]

20 Jun 2020

Please see the attached response to reviewers that has been uploaded as a part of this revision.

---

## [Editor Report · Decision Letter 1]

26 Jun 2020

Using machine learning to predict risk of incident opioid use disorder among fee-for-service Medicare beneficiaries: a prognostic study

PONE-D-20-09504R1

Dear Dr. Lo-Ciganic,

We’re pleased to inform you that your manuscript has been judged scientifically suitable for publication and will be formally accepted for publication once it meets all outstanding technical requirements.

Kind regards,

Kevin Lu, PhD

Academic Editor

PLOS ONE

---

## [Editor Report · Acceptance letter]

6 Jul 2020

PONE-D-20-09504R1 

Using machine learning to predict risk of incident opioid use disorder among fee-for-service Medicare beneficiaries: a prognostic study 

Dear Dr. Lo-Ciganic:

I'm pleased to inform you that your manuscript has been deemed suitable for publication in PLOS ONE. Congratulations! Your manuscript is now with our production department. 

Kind regards, 

on behalf of

Professor Kevin Lu 

Academic Editor

PLOS ONE